# The Association of Prostate Cancer and Urinary Tract Infections: A New Perspective of Prostate Cancer Pathogenesis

**DOI:** 10.3390/medicina59030483

**Published:** 2023-03-01

**Authors:** Szu-Ying Pan, Wen-Chi Chen, Chi-Ping Huang, Chung Y. Hsu, Yi-Huei Chang

**Affiliations:** 1Department of Urology, China Medical University Hospital, No. 2, Yuh-Der Road, North District, Taichung 404332, Taiwan; 2Graduate Institute of Integrated Medicine, College of Chinese Medicine, China Medical University, Taichung 40402, Taiwan; 3School of Medicine, College of Medicine, China Medical University, Taichung 40402, Taiwan; 4Graduate Institute of Biomedical Sciences, China Medical University, Taichung 404327, Taiwan; 5College of Public Health, China Medical University, Taichung 406040, Taiwan

**Keywords:** prostate cancer, urinary tract infection, pyelonephritis, cystitis, prostatitis, inflammation, microbiota, microbiome

## Abstract

*Background and objectives:* Microbiota of the urinary tract may be associated with urinary tract malignancy, including prostate cancer. *Materials and Methods:* We retrospectively collected patients with newly diagnosed prostate cancer and subjects without prostate cancer from the National Health Insurance Research Database (NHIRD) in Taiwan between 1 January 2000 and 31 December 2016. A total of 5510 subjects were recruited and followed until the diagnosis of a primary outcome (urinary tract infection, pyelonephritis, cystitis, and prostatitis). *Results*: We found that the patients with prostate cancer had a significantly higher risk of urinary tract infections than those without prostate cancer. The adjusted hazard ratios for pyelonephritis, prostatitis, and cystitis were 2.30 (95% CI = 1.36–3.88), 2.04 (95% CI = 1.03–4.05), and 4.02 (95 % CI = 2.11–7.66), respectively. We clearly identified the sites of infection and associated comorbidities in the prostate cancer patients with urinary tract infections. In addition, we found that the patients receiving radiotherapy and androgen deprivation therapy had a lower risk of urinary tract infections than the patients in corresponding control groups. *Conclusions*: Our study suggests that an abnormal urine microbiome could potentially contribute to the development of prostate cancer through inflammation and immune dysregulation. Furthermore, an imbalanced microbiome may facilitate bacterial overgrowth in urine, leading to urinary tract infections. These findings have important implications for the diagnosis and treatment of prostate cancer. Further research is needed to better understand the role of the urine microbiome in prostate cancer pathogenesis and to identify potential microbiome-targeted therapies for the prevention and treatment of prostate cancer.

## 1. Introduction

Prostate cancer is the most common genitourinary tract malignancy and the second most common cancer in men [1]. According to the Global Cancer Observatory (GLOBOCAN) study, there were 1,276,106 newly diagnosed cases of prostate cancer in 2018 [2]. However, this number had increased to 1,414,259 by 2020 [3]. Risk factors associated with the formation of prostate cancer include androgen, genetics, diet, smoking, family history, and ethnicity [4]. However, there is no strong evidence of effective preventive methods for prostate cancer [1], and, therefore, studying factors associated with prostate cancer remains important.

Bacteria have been associated with the development of several cancer types, and in particular cancer of the genitourinary tract cancer. Possible mechanisms include inflammation or metabolic processes influenced by the microbiota [5]. Prostate cancer is the most interesting of these genitourinary tract malignancies, not only because of the diverse treatments and evolving technologies for the diagnosis [6] and management, but also because many factors remain unknown [7]. Bacteria may affect the formation and progression of malignancies through pathways, such as susceptibility, inflammatory cytokines, and host immunity [8]. The role of bacteria in the development of bladder cancer has been proposed [9], but the evidence was weak when pooling data from more high-quality reports in a review by Bayne et al. [10].

In this study, we aimed to investigate the association between urinary tract infections and prostate cancer using data from a nationwide population-based cohort database. In addition, we discuss the potential connection between urinary tract microbiota and prostate cancer.

## 2. Materials and Methods

### 2.1. Data Source

Taiwan launched the National Health Insurance (NHI) program, a compulsory social insurance program, in 1995, and it currently provides healthcare for more than 99% of the population. This study used data from the Longitudinal Generation Tracking Database (LGTD 2005), which contains the data of two million individuals randomly selected from the National Health Insurance Research Database (NHIRD). The NHIRD contains detailed information on healthcare utilization, including hospital admissions, outpatient visits, and prescription medications. In addition, diagnoses are recorded according to the International Classification of Diseases, Ninth & Tenth Revisions, Clinical Modification (ICD-9-CM and ICD-10-CM). This study was approved by the Institutional Review Board of China Medical University Hospital Research Ethics Committee (CMUH109-REC2-031(CR-2)).

### 2.2. Study Population, Primary Outcomes and Covariates

We identified patients with newly diagnosed prostate cancer in the NHIRD. The index date was defined as the date of diagnosis between 1 January 2000 and 31 December 2016. We also enrolled patients without prostate cancer as the non-prostate cancer cohort, and their index date was defined as a random date between 2000 and 2016, and the index year was the year of the index date. Both cohorts were matched at a 1:1 ratio by age and index year. We excluded female patients, male patients who were aged <20 years, and male patients who had a prior history of infection (including urinary tract infection, prostatitis, cystitis, and pyelonephritis) before the index date. We also excluded patients without data on sex and age. The primary outcomes were urinary tract infection, prostatitis, cystitis, and pyelonephritis. Comorbidities including hypertension, diabetes, hyperlipidemia, chronic obstructive pulmonary disease (COPD), and chronic kidney disease (CKD) were recorded in both cohorts. In the patients with prostate cancer, data on radiotherapy, chemical therapy, surgery (radical prostatectomy), and androgen deprivation therapy were also recorded. A total of 5510 subjects were followed until the diagnosis of a primary outcome: loss to follow-up, death, or 31 December 2017.

### 2.3. Statistical Analysis

Categorical data are presented as numbers and percentages, and continuous data are presented as means and standard errors. The patients were classified into two age groups for analysis: 20–64 years, and ≥65 years. The chi-square test and *t* test were used to analyze differences in categorical variables and continuous variables, respectively. The incidence rate of urinary tract infections (including prostatitis, cystitis, and pyelonephritis) was calculated. Univariable and multivariable Cox proportional hazard models were used to compute crude hazard ratios (HRs), adjusted HRs, and corresponding 95% confidence intervals (Cis). The multivariable Cox proportional hazard model was adjusted for age, comorbidities, medications and treatment for prostate cancer. Kaplan–Meier survival curves were used to compare the cumulative incidence between the non-prostate cancer and prostate cancer cohorts, and the log-rank test was used to examine the differences. All statistical analyses and graphs were performed using SAS version 9.4 (SAS Institute, Inc., Cary, NC, USA) and R studio (3.5.2).

## 3. Results

The characteristics of the non-prostate cancer and prostate cancer cohorts are presented in Table 1. Overall, 81.3% of the patients were over 64 years of age, and 18.7% were aged 20–64 years. We analyzed the data collected from a total of 2755 patients in the non-prostate cancer cohort and 2755 patients in the prostate cancer cohort. Compared to the patients without prostate cancer, those with prostate cancer had higher rates of hypertension (66.4% versus 58.7%), diabetes (25.9% versus 24.7%), hyperlipidemia (38.7% versus 32.9%), COPD (28.9% versus 25.8%), and CKD (15.5% versus 12.4%), respectively. In addition, in the patients with prostate cancer, surgery was the most common treatment.

Table 2 shows that the patients with prostate cancer were associated with urinary tract infections, prostatitis, cystitis, and pyelonephritis. In addition, the patients with prostate cancer had a significantly higher risk of urinary tract infections than those without prostate cancer (adjusted HR = 1.59, 95% CI = 1.42–1.77). Compared to the patients in the 20–64 age group, those over 64 years of age had a higher risk of urinary tract infections (adjusted HR = 1.71, 95% CI = 1.44–2.03). In addition, the patients with hypertension (adjusted HR = 1.26, 95% CI = 1.11–1.42), COPD (adjusted HR = 1.19, 95% CI = 1.06–1.34) and CKD (adjusted HR = 1.27, 95% CI = 1.10–1.48) had a significantly higher risk of urinary tract infections compared with the corresponding groups. In contrast, the patients with hyperlipidemia had a lower risk of urinary tract infections compared to those without hyperlipidemia (adjusted HR = 0.84, 95% CI = 0.74–0.95).

Comparisons of the incidence of urinary tract infections, prostatitis, cystitis, and pyelonephritis between the patients with and without prostate cancer are shown in Table 3. The patients with prostate cancer had a significantly higher risk of urinary tract infections (adjusted HR = 1.58, 95% CI = 1.41–1.76), prostatitis (adjusted HR = 2.04, 95% CI = 1.03–4.05), cystitis (adjusted HR = 4.02, 95% CI = 2.11–7.66), and pyelonephritis (adjusted HR = 2.30, 95% CI = 1.36–3.88) than those without prostate cancer. The Kaplan–Meier survival curves of the cumulative incidence for the different outcomes are shown in Figure 1. The cumulative incidence of urinary tract infections, prostatitis, cystitis, and pyelonephritis were significantly higher in the patients with prostate cancer compared to those without prostate cancer.

As shown in Table 4, the patients with prostate cancer had a significantly higher risk of urinary tract infections than those without prostate cancer, regardless of age and comorbidities (adjusted HRs > 1, *p*-values < 0.05). In addition, the younger patients (<65 years) were more likely to have urinary tract infections than the older groups (HR = 2.78 vs. 1.50). We further divided the patients into four groups based on follow-up time: <0.5, 0.5–1, 1–2, and ≥2 years. The results still showed that the patients with prostate cancer had a significantly higher risk of urinary tract infections than those without prostate cancer, regardless of follow-up time (adjusted HRs > 1, *p*-values < 0.05). We then examined the risk of urinary tract infections among the patients with prostate cancer according to their age, comorbidities, and treatments (Table 5). We found that the patients who received radiotherapy and androgen deprivation therapy had a significantly lower risk of urinary tract infections than those who did not receive these therapies (adjusted HRs < 1, *p*-values < 0.05). In addition, age ≥ 65 years, hypertension, diabetes, COPD, and CKD were risk factors for urinary tract infections.

## 4. Discussion

Our results demonstrated that the patients with prostate cancer were significantly associated with urinary tract infections including prostatitis, cystitis, and pyelonephritis. Older age, hypertension, COPD, and CKD were also associated with an increased risk of urinary tract infections. Statins (HMG-CoA reductase inhibitors) have been reported to reduce the occurrence of urinary tract infections, as they reduce susceptibility to bacteria [11,12,13], which corresponds with our findings. Diabetes mellitus has been associated with increased frequency, severity, and likelihood of developing urinary tract infections [14,15]. However, in both the whole cohort (Table 2) and the prostate cancer group (Table 5), diabetes mellitus had no effect on the risk of urinary tract infections. Therefore, prostate cancer seems to be a more influential factor. To address the potential confounding effect of comorbidities, we constructed a non-prostate cancer cohort that was well-matched for age, sex, index year, and comorbidities, including hypertension, diabetes, hyperlipidemia, COPD, and CKD (see Appendix A). With this newly selected non-prostate cancer, our analysis revealed that the associations between prostate cancer and urinary tract infections, prostatitis, cystitis, and pyelonephritis remained significant. These findings were consistent with the original data and are presented in Appendix A. In summary, as shown in Table 4, prostate cancer is a more crucial determinant compared with other comorbidities. More importantly, for the younger prostate cancer patients (<65 years), cancer itself led to a nearly two-fold higher risk (HR:2.78 vs. 1.50) of developing a urinary tract infection.

Previous evidence has demonstrated the vital role and relevance of inflammation in prostate carcinogenesis and tumor progression [16]. Escherichia coli and Enterococcus species have also been shown to be the main bacteria associated with prostate inflammation. In recent years, many studies have investigated the role of the microbiome as a biomarker of disease [17]. The potential relevance of the urinary tract microbiome on prostate cancer diagnosis and treatment has also attracted attention, and many studies have investigated this relationship. The findings of these studies have suggested that urine is not sterile and that micro-organisms represent a distinct flora in the urinary tract. In addition, differences in the microbiome between patients with and without urinary tract infections have been reported, suggesting that the microbiome may have an effect on the susceptibility to infection in prostate cancer patients [18]. With emerging technologies such as next-generation sequencing, microbiome signatures in patients with prostate cancer have been described [19]. Yu et al. investigated the microbiota in expressed prostate secretion (EPS) from patients with prostate cancer, and found a significantly increased number of bacteria than in patients with benign prostate hyperplasia [20]. Interestingly, the number of E-coli was decreased in urine but increased in EPS and seminal fluid. This may also imply that a bacteriological change plays a role in prostate cancer. Moreover, recent research has identified five types of bacteria (Anaerococcus, Peptoniphilus, Porphyromonas, Fenollaria, and Fusobacterium) that are linked to a more rapid progression of prostate cancer to an aggressive state [21]. Good control of the disease can also decrease the risk of urinary tract infections.

Our findings also revealed that radiation therapy was a protective factor against urinary tract infections in the patients with prostate cancer. Tolani et al. analyzed 118 Nigerian patients with prostate disease of whom 22% had prostate cancer, and found that the only risk factor for acute urinary tract infections was an indwelling catheter [22]. In addition, Tunio et al. reported that 16.6% of localized prostate cancer patients under curative radiotherapy had positive urine cultures [23]. However, all of their patients had indwelling catheters for at least 4 weeks, and therefore the cause of urinary tract infections in these two series may be related to the placement of catheters. We did not find this association. Furthermore, irradiation is thought to cause local immunosuppression by inducing tumor-associated macrophages, myeloid-derived suppressor cells, and regulatory T cells [24]. We hypothesize that these processes lead to conditions of immunosuppression in the micro-environment, and this reduces inflammation in the prostate, further upregulating the urine microbiome. Changing the micro-organisms in urine may make it less likely for bacteria to cause urinary tract infections.

Androgen deprivation therapy was shown to significantly reduce total prostate volume by 37.7% and improve lower urinary tract symptoms in patients with prostate cancer [25,26]. Improved voiding can reduce the risk of urinary tract infections. Patients in the radical prostatectomy group also had a lower incidence of urinary tract infections, but the difference was not statistically significant. This may be because patients undergoing radical prostatectomy may receive post-operative antibiotics, and removal of the prostate gland may also improve voiding. However, the urine microorganisms were still unchanged. Androgen ablation results in hypoxia [27]. Intraprostatic hypoxia has been associated with the early recurrence of prostate cancer [28]. Recent research has identified five types of mostly anaerobic bacteria that are linked to a more rapid progression of prostate cancer. That is, androgen deprivation therapy can reduce the risk of urinary tract infection by the shrinkage of prostate volume, treating prostate cancer by blocking the hypothalamic-pituitary-gonadal axis [29], but its biochemical mechanism is associated with early disease recurrence and rapid progression.

Patients with advanced prostate cancer may experience bladder outlet obstruction, which can cause urine reflux and increase the risk of upper urinary tract infections, such as cystitis and pyelonephritis. Our study showed that prostate cancer patients are more susceptible to urinary tract infections, including these types of infections, compared to the general population. These findings highlight the importance of close monitoring and early intervention for urinary tract infections in prostate cancer patients.

In our study, patients without prostate cancer were confirmed to have undergone prostate biopsy, which is a common diagnostic procedure that can cause urinary tract infections. Previous studies have shown that fluoroquinolones are an effective antibiotic prophylaxis for patients undergoing prostate biopsy, particularly in males with diabetes mellitus, who are at higher risk for urinary tract infections [30]. In our cohorts, antibiotics are routinely prescribed to patients, with fluoroquinolones being the most commonly used antibiotic. A systematic review regarding prostate cancer synchronous/metachronous with colorectal cancer by Celentano et al. reported that it was probably due to a combination of genetic and environmental factors [31]. Due to prostate cancer frequently occurring in older men, sharing same environment may have contributed to it being synchronous with rectal cancer. Bacteria inoculation in the rectum and urethra may be the same species, but the evidence remains weak. Our results may contribute to this hypothesis.

Our study has several limitations that should be considered. Firstly, we used data from a nationwide database that did not include information on laboratory findings, staging, possible risk factors, or the treatment course. Additionally, we lacked data on the patients’ personal histories, such as smoking or alcohol consumption. Secondly, the small number of patients and the predominantly Asian population are limitations of this study. Furthermore, the use of the International Classification of Diseases code to identify disease events may have resulted in an underestimation of the number of events for prostatitis, cystitis, and pyelonephritis. Despite these limitations, our findings revealed a tendency for urinary tract infections to affect both the upper and lower urinary tracts.

## 5. Conclusions

In summary, this study used nationwide population data to investigate the relationship between prostate cancer and urinary tract infections. The study not only explored the connection between prostate cancer and urinary tract microbiota but also identified associated comorbidities and types of infection. The findings suggest that androgen deprivation and radiation therapy may impact the pathogenesis and prognosis of prostate cancer, which challenges the current belief that surgery and radiation plus androgen deprivation therapy are of equal importance. Additionally, prostate cancer is an influential factor associated with the risk of urinary tract infections, particularly in younger patients. Beyond cancer, anti-inflammatory agents may also help prevent urinary tract infections by altering the prostate microenvironment. Contrary to common belief, our findings suggest that urinary tract infections do not cause prostate cancer. Rather, an abnormal urine microbiome may increase the risk of urinary tract infections, which are associated with prostate cancer. Additionally, our study sheds light on the relationship between prostate cancer treatments and the occurrence of urinary tract infections. These findings have important implications for improving patients’ quality of life and call for further research in this area.

## Figures and Tables

**Figure 1 medicina-59-00483-f001:**
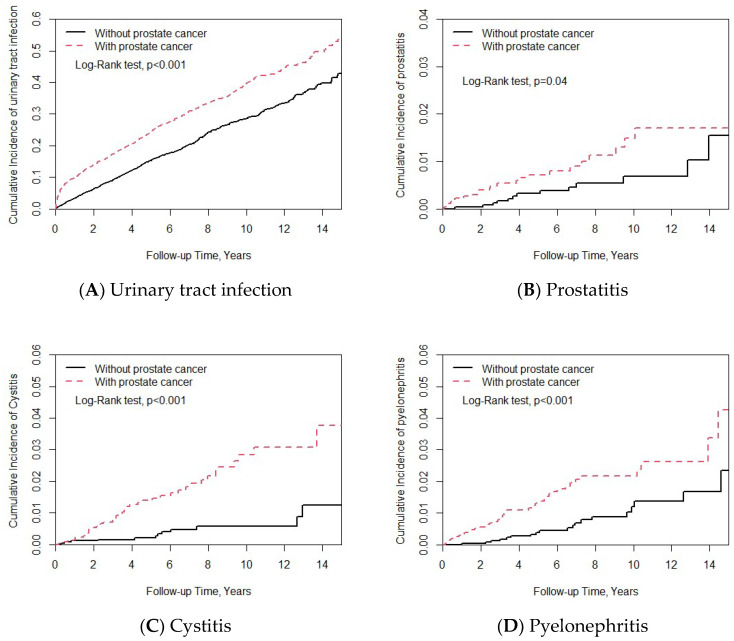
Cumulative incidence of (**A**) urinary tract infections, (**B**) prostatitis, (**C**) cystitis, and (**D**) pyelonephritis between the patients with and without prostate cancer.

**Table 1 medicina-59-00483-t001:** Baseline characteristics of the patients with or without prostate cancer.

Variables	Without Prostate Cancer n = 2755		With Prostate Cancer n = 2755		
	n	%	n	%	*p*-Value
Age, years					0.99
20–64	514	18.7	514	18.7	
≥65	2241	81.3	2241	81.3	
Mean ± SD ^a^					
Comorbidity					
Hypertension	1617	58.7	1829	66.4	<0.001
Diabetes	680	24.7	714	25.9	0.29
Hyperlipidemia	907	32.9	1065	38.7	<0.001
COPD	712	25.8	797	28.9	0.01
CKD	341	12.4	428	15.5	<0.001
Treatments					
Radiotherapy			1106	40.2	
Chemical therapy			361	13.1	
Radical prostatectomy			2061	74.8	
Androgen deprivation therapy			1215	44.1	

^a^ *t*-test; Chi-square test. Abbreviations: COPD, chronic obstructive pulmonary disease; CKD, chronic kidney disease.

**Table 2 medicina-59-00483-t002:** Cox proportional hazard model analysis for the outcomes (including urinary tract infections, prostatitis, cystitis, and pyelonephritis) in the patients with and without prostate cancer.

Characteristics	Event No. (n = 1349)	IR	Crude		Adjusted	
			HR (95% CI)	*p* Value	HR (95% CI)	*p* Value
Prostate cancer						
No	569	34.8	Ref.		Ref.	
Yes	780	57.9	1.64(1.47, 1.83)	<0.001	1.59(1.42, 1.77)	<0.001
Age, years						
20–64	153	26.0	Ref		Ref	
>64	1196	50.0	1.91(1.61, 2.26)	<0.001	1.71(1.44, 2.03)	<0.001
Comorbidities						
Hypertension	902	51.9	1.43(1.27, 1.60)	<0.001	1.26(1.11, 1.42)	<0.001
Diabetes	341	50.8	1.15(1.01, 1.30)	0.02	1.096(0.96,1.24)	0.20
Hyperlipidemia	433	44.7	0.97(0.87, 1.09)	0.61	0.84(0.74, 0.95)	0.004
COPD	425	57.4	1.38(1.23, 1.55)	<0.001	1.19(1.06, 1.34)	0.003
CKD	220	65.0	1.49(1.29, 1.72)	<0.001	1.27(1.10, 1.48)	<0.001

Abbreviations: IR, incidence rate; HR, hazard ratio; CI, confidence interval; COPD, chronic obstructive pulmonary disease; CKD, chronic kidney disease. Adjusted HR: adjusted for age, sex, and comorbidities in Cox proportional hazards regression.

**Table 3 medicina-59-00483-t003:** Comparisons of the incidence of urinary tract infections, prostatitis, cystitis, and pyelonephritis between the patients with and without prostate cancer.

	Without Prostate Cancer			With Prostate Cancer			Crude		Adjusted	
Variable	Event	Person-Year	IR	Event	Person-Year	IR	HR (95% CI)	*p*-Value	HR (95% CI)	*p*-Value
Urinary tract infection	557	16,415	33.9	755	13,554	55.7	1.62(1.45, 1.80)	<0.001	1.58(1.41, 1.76)	<0.001
Prostatitis	13	18,605	0.70	23	16,147	1.42	2.04(1.03, 4.03)	0.04	2.04(1.03, 4.05)	0.04
Cystitis	12	18,604	0.65	43	16,027	2.68	4.18(2.20, 7.94)	<0.001	4.02(2.11, 7.66)	<0.001
Pyelonephritis	21	18,571	1.13	44	16,140	2.73	2.45(1.46, 4.13)	<0.001	2.30(1.36, 3.88)	0.002

Abbreviations: IR, incidence rate, per 1000 person-years; HR, hazard ratio; CI, confidence interval. Adjusted HR: adjusted for age, sex, and comorbidities in Cox proportional hazards regression.

**Table 4 medicina-59-00483-t004:** Comparisons of the incidence of urinary tract infections between the patients with and without prostate cancer according to baseline characteristics and follow-up period.

	Without Prostate Cancer			With Prostate Cancer			Crude		Adjusted	
Variable	Event	Person-Years	IR	Event	Person-Years	IR	HR (95% CI)	*p*-Value	HR (95% CI)	*p*-Value
Age (years)										
20–64	46	3284	14.0	107	2603	41.1	2.81(1.99, 3.98)	<0.001	2.78(1.98, 3.94)	<0.001
≥65	523	13,075	40.0	673	10,860	62.0	1.53(1.37, 1.72)	<0.001	1.50(1.33, 1.68)	<0.001
Comorbidities										
No	132	5246	25.2	139	2841	48.9	1.93(1.52, 2.44)	<0.001	1.89(1.49, 2.40)	<0.001
Yes	437	11,112	39.3	641	10,621	60.4	1.52(1.34, 1.71)	<0.001	1.55(1.37, 1.75)	<0.001
Follow-up time (years)										
<0.5	54	1358	39.8	225	1287	174.8	4.35(3.23, 5.86)	<0.001	4.29(3.18, 5.78)	<0.001
0.5–1	38	1319	28.8	54	1209	44.7	1.55(1.02, 2.35)	0.04	1.54(1.02, 2.34)	0.04
1–2	79	2445	32.3	101	2151	47.0	1.47(1.09, 1.97)	0.01	1.39(1.04, 1.87)	0.03
≥2	398	11,236	35.4	400	8815	45.4	1.29(1.12, 1.48)	<0.001	1.26(1.10, 1.45)	0.001

Abbreviations: IR, incidence rate, per 1000 person-years; HR, hazard ratio; CI, confidence interval. Adjusted HR: adjusted for age, sex, and comorbidities in Cox proportional hazards regression.

**Table 5 medicina-59-00483-t005:** The associations between urinary tract infections and different treatments and covariates in the patients with prostate cancer.

Characteristics	Event No.	IR	Crude		Adjusted	
	(n = 780)		HR (95% CI)	*p* Value	HR (95% CI)	*p* Value
Radiotherapy						
No	513	66.6	Ref.		Ref.	
Yes	267	46.4	0.70(0.61, 0.82)	<0.001	0.71(0.61, 0.83)	<0.001
Chemical therapy						
No	691	59.8	Ref.		Ref.	
Yes	89	46.7	0.78(0.63, 0.98)	0.03	0.94(0.74, 1.19)	0.61
Radical prostatectomy for Prostate cancer						
No	203	59.1	Ref.		Ref.	
Yes	577	57.6	0.98(0.83, 1.14)	0.76	0.97(0.82, 1.14)	0.69
Androgen deprivation therapy						
No	501	62.1	Ref.		Ref.	
Yes	279	51.8	0.80(0.69, 0.93)	0.003	0.85(0.74, 0.99)	0.04
Age, years						
20–64	107	41.1	Ref.		Ref.	
>64	673	62.0	1.50(1.22, 1.84)	<0.001	1.35(1.09, 1.66)	0.006
Comorbidities						
Hypertension	541	63.3	1.28(1.10, 1.49)	0.002	1.23(1.05, 1.45)	0.01
Diabetes	197	61.8	1.06(0.90, 1.25)	0.46	1.09(0.92, 1.29)	0.34
Hyperlipidemia	259	52.1	0.84(0.72, 0.97)	0.02	0.78(0.67, 0.91)	0.002
COPD	255	69.5	1.28(1.11, 1.49)	0.001	1.16(1.00, 1.36)	0.05
CKD	139	73.7	1.31(1.09, 1.57)	0.005	1.23(1.02, 1.48)	0.03

Abbreviations: HR, hazard ratio; CI, confidence interval; COPD, chronic obstructive pulmonary disease; CKD, chronic kidney disease. Adjusted HR: adjusted for age, sex, and comorbidities in Cox proportional hazards regression.

## Data Availability

The dataset used in this study is held by the Taiwan Ministry of Health and Welfare (MOHW). The Ministry of Health and Welfare must approve our application to access this data. Any researcher interested in accessing this dataset can submit an application form to the Ministry of Health and Welfare requesting access. Please contact the staff of MOHW (Email: stcarolwu@mohw.gov.tw) for further assistance. Taiwan Ministry of Health and Welfare Address: No.488, Sec. 6, Zhongxiao E. Rd., Nangang Dist., Taipei City 115, Taiwan (R.O.C.). Phone: +886-2-8590-6848.

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
