# Peer review of "The Association of Prostate Cancer and Urinary Tract Infections: A New Perspective of Prostate Cancer Pathogenesis"

_medicina, 2023, doi:10.3390/medicina59030483_

Round 1

Reviewer 1 Report

Interesting manuscript about the association between infectious disease and PCa.

Another point of view could be represented by the antibiotics resistance spectrum and related-induced resistances and/or alteration to urinary microbiota and PCa. The authors should consider to briefly add a separated paragraph in the Discussion section. In this regard, the authors should consider to refer to a recent paper exploring the antibiotics adoption in between the EMA warning about the fluoroquinolones (doi: 10.1007/s00345-022-04055-7 ) as this class of drug were commonly adopted by urological community. Any thougths of the authors about alteration of microbiota induced by resistance spectrum?

Author Response

I have added a paragraph with advised citation about antibiotic used for patient underwent prostate biopsy(with or without prostate cancer). Most of our given regimen is ciprofloxacin/ levofloxacin. Because patients without prostate cancer also underwent biopsy with prophylactic antibiotics, hence this confounding factor about drug resistance could be ignored.
Thank you very much for precious suggestions.

Reviewer 2 Report

The manuscript by Szu-Ying Pan et al. investigates whether prostate cancer is associated with urinary tract infections (UTI) aiming to delve into prostate cancer pathogenesis. What the authors suggest in their study is that microbiome may contribute to tumor development through inflammations and immunity and as such may lead into a new in prostatece cancer diagnosis and treatment. 

However, what they have found in their study is that patients that have been diagnosed with prostate cancer and have received some form of treatment are more likely to suffer from some form of UTI later in their lifes than patients that haven’t been diagnosed. What they have shown from their study is that prostate cancer diagnosis and treatment can be a risk factor for UTIs irrespective of the treatment. This is actually anticipated as manipulations of the urinary tract system of patients with prostate cancer diagnosis can actually make them susceptible to UTIs.  However, there is nothing to support in their study results that UTIs can play a role in prostate cancer pathogenesis. Indeed, they can have a role but this is not something that comes as a result from this study. Research was not conducted correctly. 

A completely different methodology would be needed – for example they would have to include people with recurrent UTIs and then match them with UTI-free people in order to investigate whether UTI can be a risk factor for prostate cancer development, taking into consideration the many confounding factors that need to be matched.

Moreover in the introduction it is mentioned that a risk factor for prostate cancer is height which is not well supported in the literature in order to be listed among the other risk factors and should be deleted. 

Language editing is needed in order to improve the flow, especially in the discussion section. 

Reviewer 3 Report

General comments

In the present study, the author investigated the association between prostate cancer and urinary tract infections using nationwide population data. It is interesting to note that a large number of cases were reviewed and showed an increased risk of urinary tract infections in prostate cancer patients. However, we believe there are multiple problems with this paper, and it will be difficult to accept it if these problems are not resolved.

Specific comments:

1.                 The authors evaluated urinary tract infections with unknown focus separately from prostatitis, cystitis, and pyelonephritis, but urinary tract infections with unknown focus accounted for about 90% of the cases. In addition, about 75% of prostate cancer patients underwent radical total prostatectomy, which is considered an incorrect cohort for tracking the development of prostatitis. The number of events for prostatitis, cystitis, and pyelonephritis is low, and all should probably be integrated into urinary tract infections for evaluation.

2.                 In Figure 1, urinary tract infections increase in the prostate cancer group immediately after the diagnosis of prostate cancer. Prostate biopsy may be involved. In addition, about 75% underwent radical total prostatectomy, and it is possible that perioperative urinary catheterization and incontinent conditions may contribute to urinary tract infections. If so, it is possible that the prostate cancer diagnosis and treatment procedure itself is a risk for urinary tract infections rather than the prostate cancer being a risk for urinary tract infections. The authors need to consider these issues.

3.                 The authors state that radiation therapy is a protective factor against urinary tract infections in prostate cancer patients, but they also state that radiation therapy leads to an immunosuppressed state. This seems to be a contradiction in terms, since immunosuppression may reduce inflammation, but it may also promote infection. Also, dysuria due to radiation therapy could be a risk for urinary tract infection. The consideration needs to be changed.

4.                 As stated by the authors, androgen deprivation therapy may reduce the risk of urinary tract infections because it improves lower urinary tract symptoms. Radiation therapy may be used in conjunction with androgen deprivation therapy, and it is necessary to add how many prostate cancer patients who received radiation therapy also received androgen deprivation therapy.

5.                 Line 212, 'Table 4 shows that … .''

The authors state that there are fewer urinary tract infections in the first 6 months, but Table 4 shows a higher incidence of urinary tract infections within 6 months in the prostate cancer group, which is inconsistent. Also, at the 2-year follow-up, urinary tract infections are assumed to increase with disease progression, but many cases received curative treatment, and it is estimated that not many cases of prostate cancer progressed. The authors need to revise their considerations.

6.                 The authors state that psychological influences in prostate cancer patients may be related to urinary tract infections, but the discussion is insufficient. We know that there is a negative psychological impact on cancer patients, but we do not know if it is directly related to the occurrence of urinary tract infections. Since negative psychological impact is also present in other types of cancer, it is difficult to link prostate cancer and urinary tract infections with this theory.

7.                 PI-RADS and PHI are not relevant to this study and need not be discussed in the discussion.

Round 2

Reviewer 2 Report

The manuscript has been improved greatly according to previous suggestions.

Reviewer 3 Report

I believe the article has been improved in response to reviewer comments.